# Mutational landscape of normal epithelial cells in Lynch Syndrome patients

Bernard C. H. Lee[1], Philip S. Robinson [2,3], Tim H. H. Coorens [2], Helen H. N. Yan [1], Sigurgeir Olafsson[2], Henry Lee-Six[2], Mathijs A. Sanders[2], Hoi Cheong Siu[1], James Hewinson[2], Sarah S. K. Yue[1], Wai Yin Tsui[1], Annie S. Y. Chan[1], Anthony K. W. Chan[1], Siu Lun Ho[1], Peter J. Campbell [2], Inigo Martincorena [2], Simon J. A. Buczacki[4], Siu Tsan Yuen[1], Suet Yi Leung [1,5✉] & Michael R. Stratton [2✉]

Lynch Syndrome (LS) is an autosomal dominant disease conferring a high risk of colorectal cancer due to germline heterozygous mutations in a DNA mismatch repair (MMR) gene. Although cancers in LS patients show elevated somatic mutation burdens, information on mutation rates in normal tissues and understanding of the trajectory from normal to cancer cell is limited. Here we whole genome sequence 152 crypts from normal and neoplastic epithelial tissues from 10 LS patients. In normal tissues the repertoire of mutational processes and mutation rates is similar to that found in wild type individuals. A morphologically normal colonic crypt with an increased mutation burden and MMR deficiency-associated mutational signatures is identified, which may represent a very early stage of LS pathogenesis. Phylogenetic trees of tumour crypts indicate that the most recent ancestor cell of each tumour is already MMR deficient and has experienced multiple cycles of clonal evolution. This study demonstrates the genomic stability of epithelial cells with heterozygous germline MMR gene mutations and highlights important differences in the pathogenesis of LS from other colorectal cancer predisposition syndromes.

[1] Hereditary Gastrointestinal Cancer Genetic Diagnosis Laboratory, Department of Pathology, School of Clinical Medicine, The University of Hong Kong, Queen Mary Hospital, Pokfulam, Hong Kong. [2] Wellcome Sanger Institute, Hinxton CB10 1SA, UK. [3] Department of Paediatrics, University of Cambridge, Cambridge CB2 0QQ, UK. [4] Nuffield Department of Surgical Sciences, University of Oxford, Headington, Oxford OX3 7DQ, UK. [5] Centre for PanorOmic Sciences, The University of Hong Kong, Pokfulam, Hong Kong. ✉email: suetyi@hku.hk; mrs@sanger.ac.uk

Lynch Syndrome (LS) is the most common cause of hereditary colorectal cancer (CRC), accounting for ~3% of all CRC cases[1]. LS patients carrying a germline mutation in one of the MMR genes, typically *MLH1*, *MSH2*, or *EPCAM* deletion, have a lifetime CRC risk of 41–80% and endometrial cancer risk of 40–60%, with median age of onset in the 40s[2–5]. Cancer onset is thought to be a result of inactivation of the remaining functional MMR gene allele, which leads to malfunction of the MMR system. Failure to correct mismatched bases and errors from DNA slippage during replication leads to hypermutation and microsatellite instability (MSI), increasing the chance of acquiring cancer driver mutations and thus the development of adenomas and cancers.

A recent study of individuals with germline *POLE* or *POLD1* exonuclease domain mutations and predisposition to colorectal neoplasia showed that somatic mutation rates were elevated in all cells in all normal tissues examined, with the increased burden being attributed to distinctive mutational signatures[6]. Both POL and MMR proteins are involved in DNA repair during replication, and their loss of function can lead to hypermutation in tumours[7]. However, whether normal tissues in LS display any generalized increase in somatic mutation rate is uncertain[8,9] as is the frequency and timing of further elevations in mutation rates, including those associated with wild-type allele loss or neoplastic change.

In this study, we show that the mutation rates of normal epithelium cells in LS patients are similar to wild-type individuals. We identify a morphologically normal crypt displaying increased mutation burden and mutational signatures consistent with MMR deficiency, potentially representing the earliest stage in LS tumourigenesis. Phylogenetic tree reconstructions demonstrate that the most recent ancestor crypt in LS cancers has accumulated many mutations, including key cancer drivers, before the final clonal expansion. These findings demonstrate the genomic stability of normal tissues in LS patients, whose mutation accumulation patterns during tumour development highlight the unique differences of LS tumourigenesis from other CRC predisposition syndromes.

## Results

**Mutation rates of normal LS epithelium are not elevated.** 10 LS patients aged 23–78 were recruited to the study. From normal tissues, intestinal crypts (93 colorectal, 14 small intestine) were laser capture microdissected from eight individuals, endometrial glands from one individual, and gastric glands from one individual. Microscopically completely normal crypts were randomly selected for dissection ("survey crypts") (Fig. 1a). However, given the theoretical possibility that just a subset of non-neoplastic crypts might have become MMR deficient, an additional series was selected on the basis of specific morphologies including budding, branching, and unusual shapes. Furthermore, we immunohistochemically stained normal tissue from 9 LS patients with antibodies to MLH1, MSH2, or PMS2 to find individual crypts which were negative and which may, therefore, have already lost DNA MMR functions. Crypts were also sampled from three colonic adenocarcinomas from three patients, with one including an adenomatous component at the tumour edge. A total of 132 non-neoplastic crypts/glands including 107 intestinal crypts, 8 endometrial glands, 17 gastric glands, and 20 colorectal neoplasm crypts were individually microdissected and whole genome sequenced to ~33X coverage (Supplementary Fig. 1 and Supplementary Data 1).

To assess if the microdissected crypts each contained cells originating from a single adult stem cell, the variant allele frequency (VAF) distribution was assessed for each crypt.

All crypts showed a VAF peak around 0.5 and a median VAF > 0.3, confirming their clonal nature (Supplementary Fig. 2). The median base substitution mutation rates in morphologically normal crypts/glands (i.e., "survey" crypts) from the stomach, the small intestine, the colon, and the endometrium were 38.5, 72.6, 55.2, and 35.9 substitutions per year, respectively, while the median single base insertion and deletion (ID) rates were 1.6, 2.8, 2.6 and 2.1 per year, respectively (Fig. 1b). Colonic crypts with specific morphologies, including budding, branching, and unusual shapes, and also normal crypts adjacent to the edge of a tumour, showed comparable SBS and ID rates to survey crypts, and the presence of driver mutations was not associated with a particular crypt type (Fig. 1c, d). To test if the mutation rate of normal colonic crypts from LS patients is different from that of wild type individuals, we combined the LS dataset with data from 424 crypts from 42 individuals sequenced as part of our earlier study of the mutation landscape of the normal colon (Supplementary Data 2)[10,11]. We used a linear mixed-effect model to test for differences in the mutation rate between LS and wild type crypts. Likelihood ratio tests showed that including a fixed-effect variable for LS status did not improve the fit of the model (SBS: $p = 0.11$, ID: $p = 0.14$), suggesting that if there are differences between the mutation rate of wild type and LS crypts, they are too subtle to be detected in this study. There were, however, specific groups of crypts with modestly elevated mutation burdens, including those from the small intestine of PD43853, who had received prior chemotherapy treatment, and antral mucosal glands of PD45540 who had prior *Helicobacter pylori* infection and chemotherapy treatment (Supplementary Data 3 and Supplementary Data 4).

To confirm that the mutation rate of normal LS tissue is not generally elevated, clonal organoid cultures were established from normal gastrointestinal adult stem cells (ASCs) from two patients (PD45539 and PD45540) and whole genome sequenced (Supplementary Fig. 3a). Mutation rates over the lifetimes of these patients and mutational signatures were comparable to those of organoids from a wild type individual (Supplementary Fig. 3b, c and Supplementary Data 5). Furthermore, extended culture of LS organoids for >4 months did not reveal an increased in vitro mutation rate compared to organoids from the wild-type individual, confirming the stability of the genome in normal cells from LS patients (Supplementary Fig. 3d and Supplementary Data 6).

**A morphologically normal crypt shows early MMR deficiency.** We examined intestinal, endometrial, and gastric epithelium from 9 LS patients using immunohistochemistry for MLH1, MSH2, or PMS2 proteins. A single colonic crypt (lo0054) was found to be negative for MSH2 staining in a young patient (PD46179) with a germline *MSH2* mutation. Subsequent microdissection of this crypt and whole genome sequencing showed elevated SBS and ID rates compared to survey crypts, with the mutation rate estimated to be 197 SBSs and 37 IDs per year (Fig. 1c). The elevated mutation rates were primarily due to increased rates of SBS1 and SBS44 together with ID1 and ID2 (Fig. 1e and Supplementary Fig. 4d). It is conceivable that the elevated rate of SBS1 (a mutational signature predominantly characterized by C > T mutations at CG dinucleotides) was due to defective processing of G:T mismatches (a known function of the MMR system), and SBS44 is a signature known to be associated with defective DNA MMR system. The crypt harboured two potential driver mutations, *NF2* R172fs and *PIK3R1* L30fs. Taken together, the evidence indicates that this morphologically normal crypt already had defective DNA MMR function and thus may represent an extremely early stage of the pathogenesis of LS.

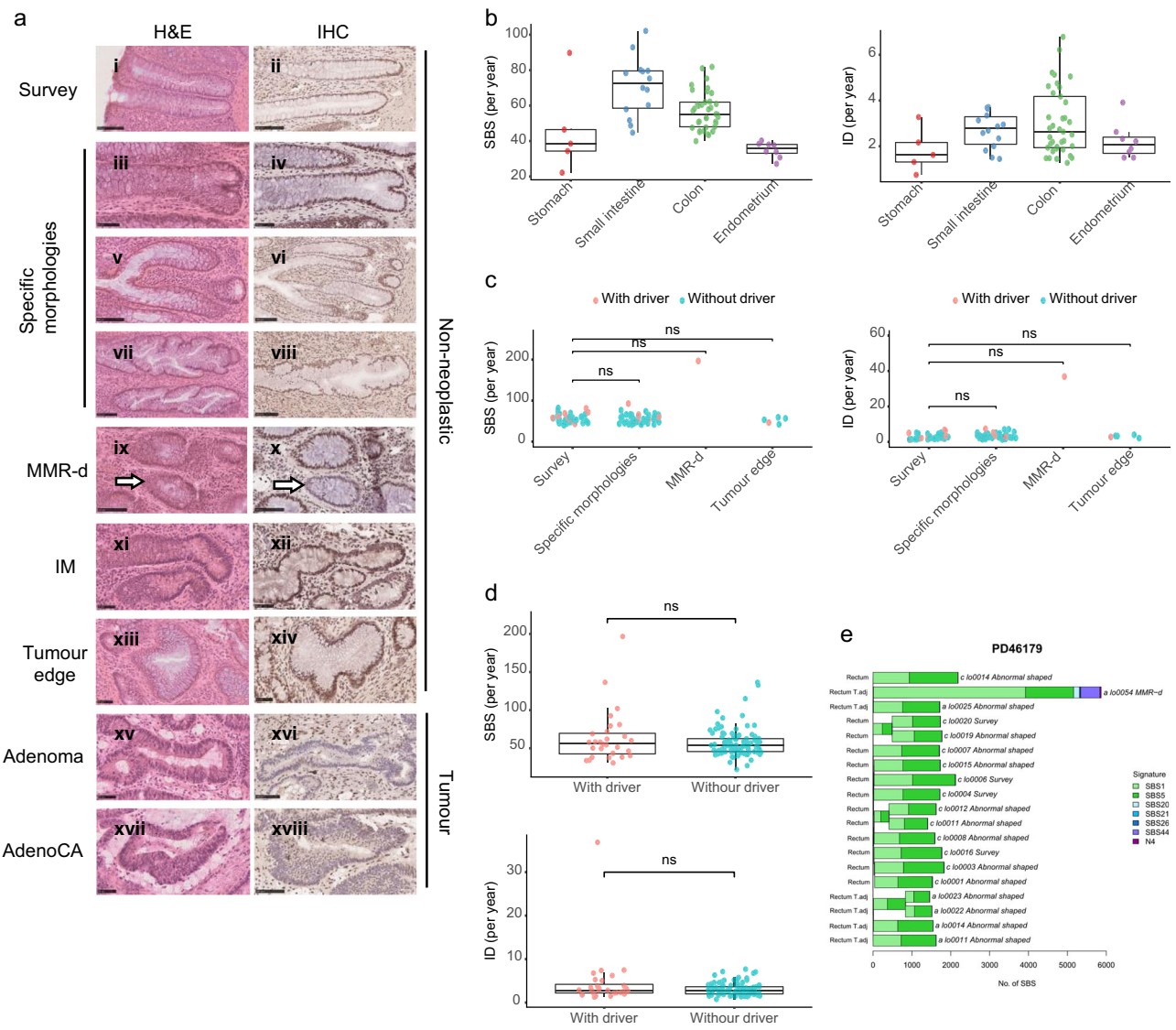

**Fig. 1 Mutation burden in the non-neoplastic crypts in LS patients. a** Representative images of H&E and IHC staining of crypts with different morphologies. Except for the MMR deficient crypt (indicated by white arrows), images of H&E and IHC did not represent the same crypt. MMR protein antibodies were used according to the germline MMR gene mutation in respective patients. MMR-d: MMR-deficient; IM: intestinal metaplasia; AdenoCA: adenocarcinoma. Scale bar: 100 μm (i, ii, v, vi, vii, viii, ix, x, xvi, xviii); 50 μm (iii, iv, xi, xii, xiii, xiv, xv, xvii). **b–d** Boxplots showing the 1.5X interquartile range (whiskers), the first and third quartile (bounds of box), and the median (centre)of the data. Each data point represented a crypt. Two-sided Wilcoxon test: ns = non-significant. **b** Single base substitutions (SBS) and insertions and deletions (ID) burden of survey crypts from the stomach ($n = 5$, red), the small intestine ($n = 14$, blue), the colon ($n = 37$, green) and the endometrium ($n = 8$, purple). **c** SBS and ID burden in different types of colonic crypts (Survey ($n = 37$), Specific morphologies (n = 50), MMR-d ($n = 1$), Tumour edge ($n = 5$)). Crypts with and without driver mutations are indicated in pink and blue, respectively. **d** SBS and ID burden in non-neoplastic crypts with ($n = 28$, pink) and without ($n = 104$, blue) cancer driver mutations. **e** A phylogenetic tree showing the clonal relationship between non-neoplastic crypts in PD46179. Branch lengths correspond to the number of SBS mutations. SBS mutational signatures are mapped onto tree branches. Each crypt is annotated with its identifier, crypt type, and tissue origin.

Interestingly, and notwithstanding a careful search, this crypt did not show a somatic *MSH2* mutation nor an LOH event that would cause inactivation of the wild type *MSH2* allele. The absence of an *MSH2* second hit, despite the elevated mutation burden and evidence of MSI, suggests that some degree of loss of MMR function and emergence of the MSI phenotype can sometimes precede inactivation of the wild type MMR gene allele in LS. The nature of the somatic event that has conferred MMR deficiency in this normal crypt is unclear. However, the observation may suggest a multistep model of progression to full MMR deficiency in LS in which normal crypts first acquire a degree of MMR deficiency, through a currently unknown mechanism, which results in an elevated SBS and ID mutation rate. In turn, this causes mutation and inactivation of the inherited, wild type MMR allele conferring a greater degree of MMR deficiency and an even higher mutation rate that leads to neoplastic changes.

**Driver mutations in normal LS epithelium.** Of the 132 glands/crypts (intestinal, gastric, and endometrial glands/crypts included) from normal epithelium, 28 (21.2%) had putative cancer driver mutations. Among these 28 glands, 23 (17.4%) had one driver mutation and five (3.8%) had two (Fig. 2a). Altogether there were 30 unique driver mutation events involving 24 cancer driver genes and all were heterozygous (Fig. 2b and

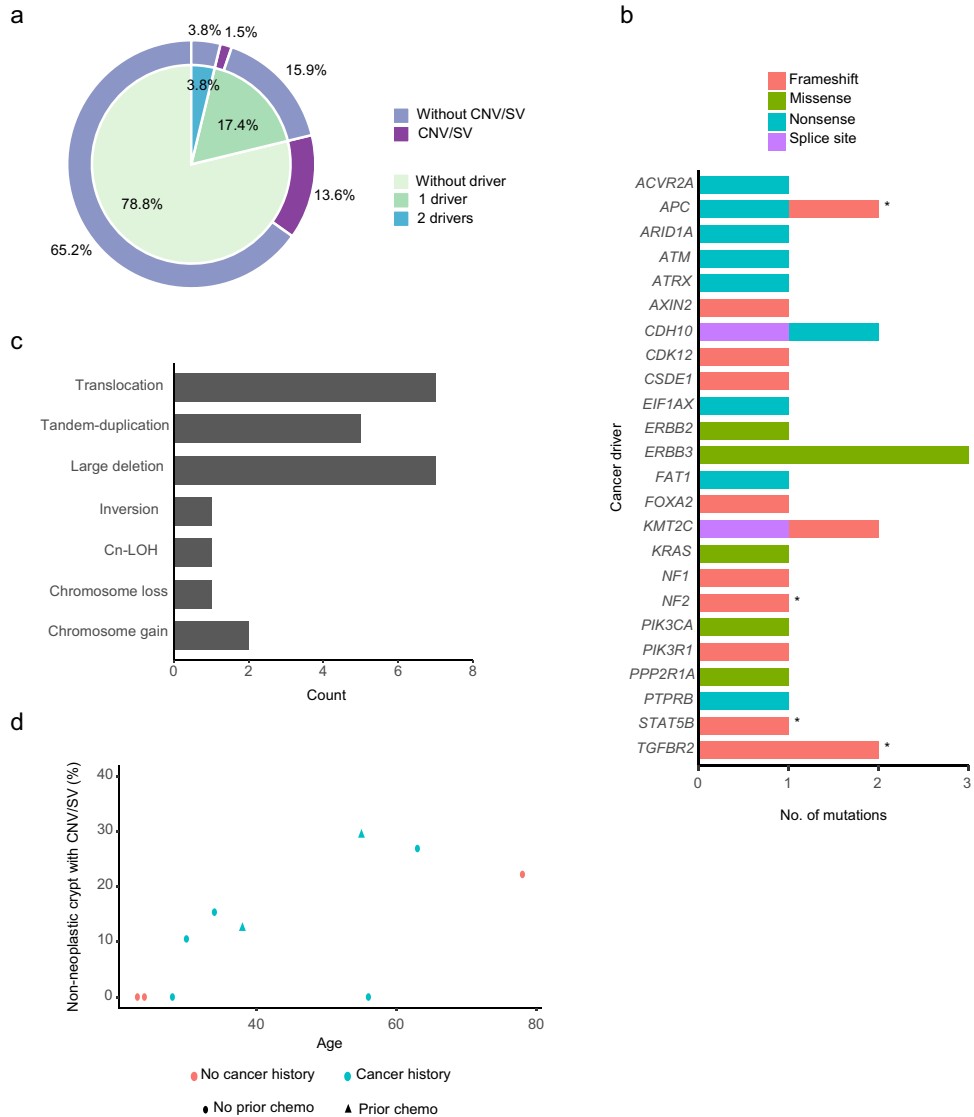

**Fig. 2 Genomic stability in the non-neoplastic crypts in LS patients. a** Distribution of non-neoplastic crypts with driver mutations and copy number or structural variants. **b** Driver mutations in non-neoplastic crypts. Asterisks denote frameshift mutations at microsatellite regions (repeat length ≥ 5). **c** Copy number variants and structural variants in non-neoplastic crypts. Cn-LOH: copy number neutral loss of heterozygosity. **d** Percentage of non-neoplastic crypts with copy number or structural variants in each LS patient.

Supplementary Data 7). Most of the cancer drivers observed in the gastrointestinal tract involved tumour suppressor genes. *APC*, the most common cancer gene mutated in CRCs, was mutated in two crypts, one (*APC* T1556fs) from the duodenum and the other (*APC* Q8X) from the caecum (Fig. 3a, b). There was no evidence of dysplastic change or difference in mutation burden in crypts carrying driver mutations. Mutations of oncogenes, including *KRAS*, *PIK3CA,* and *ERBB2*, were observed in endometrial glands, again without obvious change in morphology or mutation rate, as previously reported in wild type individuals[12] (Fig. 3c). Four copy number variants (CNVs) and 20 structural variants (SVs) were identified, affecting 15.2% of crypts (Fig. 2c), similar in frequency to wild-type individuals. There was a trend for an increasing number of CNVs/SVs with age (Fig. 2d). No homozygous deletions were observed, and the break points of structural variants did not intersect with any cancer driver genes. Interestingly, a chromosome 3p uniparental disomy (UPD) event was found in an appendix crypt (lo0002) in PD45539, a patient with a germline *MLH1* N38K mutation (Fig. 3b). This UPD event,

however, reverted the germline *MLH1* mutation back to wild type, re-endowing the crypt with two functional copies of *MLH1*.

**Mutational processes in the LS epithelium**. To investigate the genomic characteristics and mutational processes operating in the crypts of LS patients, phylogenetic trees of crypts were constructed, followed by mutational signature extraction for each patient. Mutational processes in normal epithelium from LS patients were similar to those reported in wild type individuals[10,12]. All crypts/glands from LS patients predominantly displayed the clock-like signatures SBS1 and SBS5, with SBS2, SBS13, SBS17b, SBS18, SBS35, and SBS88 also present in some crypts/glands from some patients (Fig. 3). SBS35, a signature associated with platinum-based drug usage[13], was identified in every intestinal crypt in PD43853 who had undergone a combination of 5-fluorouracil (5-FU), leu-covorin, and oxaliplatin (FOLFOX) treatment before surgery (Fig. 3a). SBS2 and SBS13, two mutational signatures associated with APOBEC enzyme activities[14,15], were observed in some

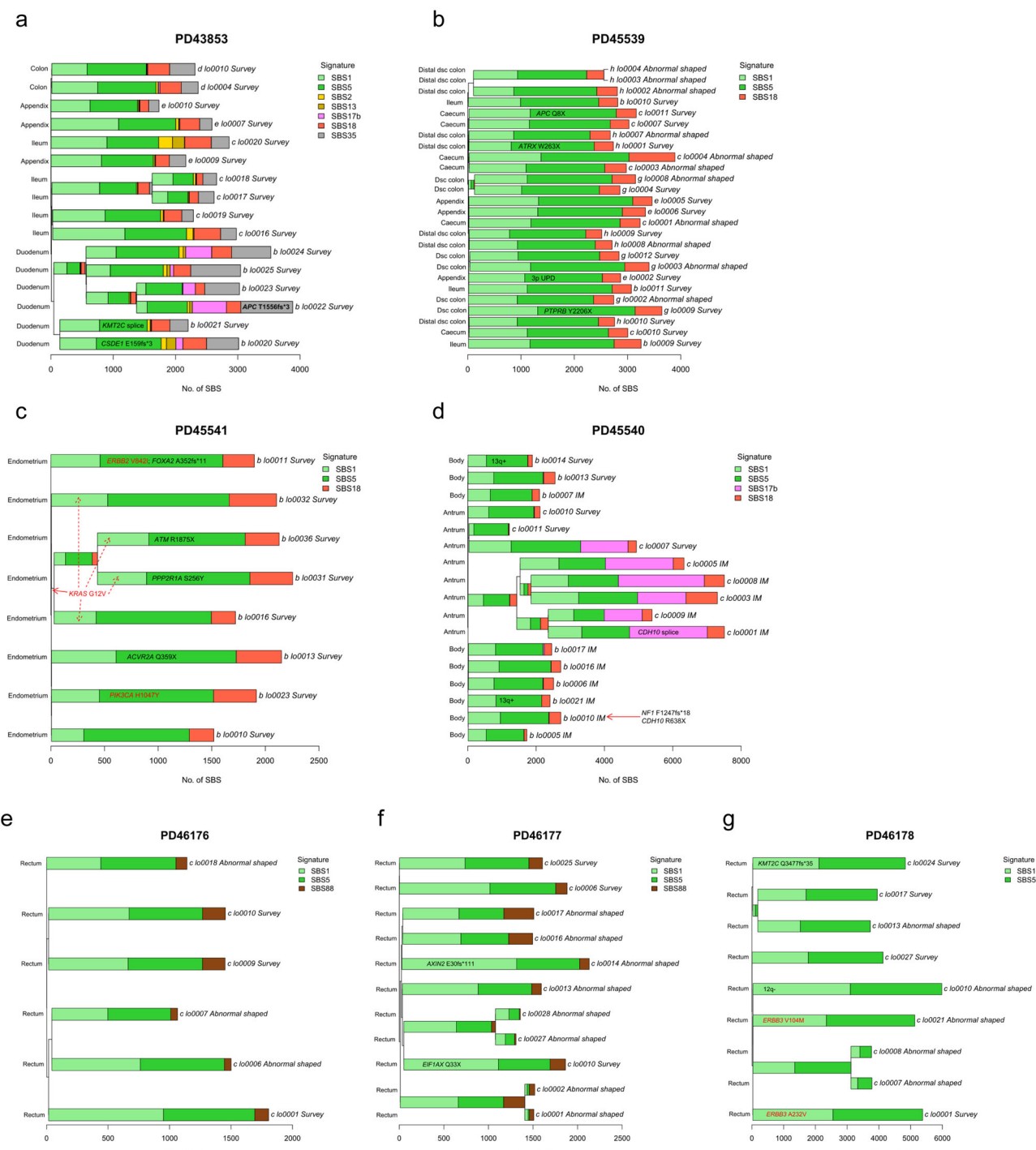

**Fig. 3 SBS mutational signatures in the normal epithelium in LS patients.** Phylogenetic trees showing the clonal relationship between non-neoplastic crypts in 7 patients (**a**, **b**, **e**, **f**, **g** intestinal; **c** endometrial; **d** gastric). Branch lengths correspond to the number of SBS mutations. SBS mutational signatures are mapped onto tree branches. Each crypt is annotated with its identifier, crypt type, and tissue origin. Cancer driver mutations and copy number variants are labelled on the corresponding tree branch. Frameshift mutations at microsatellite regions (repeat length ≥ 5) are indicated in bold. Activating mutations of oncogenes are indicated in red. Inactivating mutations of tumour suppressors are indicated in black. **c** *KRAS* G12V was an early event and thus annotated on the common branch (red arrow) of the 4 crypts harbouring this mutation (dotted red arrows).

duodenal and ileal crypts from PD43853, potentially reflecting the presence of localized DNA editing by cytidine deaminases in the small intestine (Fig. 3a). SBS17b, a sub-signature of SBS17 that has in some instances been linked to 5-FU exposure and oxidative stress environment[16], was observed in most crypts in the pyloric antrum but not the body of the stomach in PD45540 who had been treated with 5-FU (Fig. 3d). In a family with a germline *EPCAM* deletion, SBS88, a signature associated with colibactin exposure[10,17], was found in every crypt from two siblings (PD46176 and PD46177) in their early twenties, but not in their grandmother (PD46178) (Fig. 3e–g). Colibactin is a genotoxin secreted by pks+ *E.coli*, typically present in the gut microbiota[18,19]. The extensive presence of SBS88 in the two siblings suggests a shared microbiome between them.

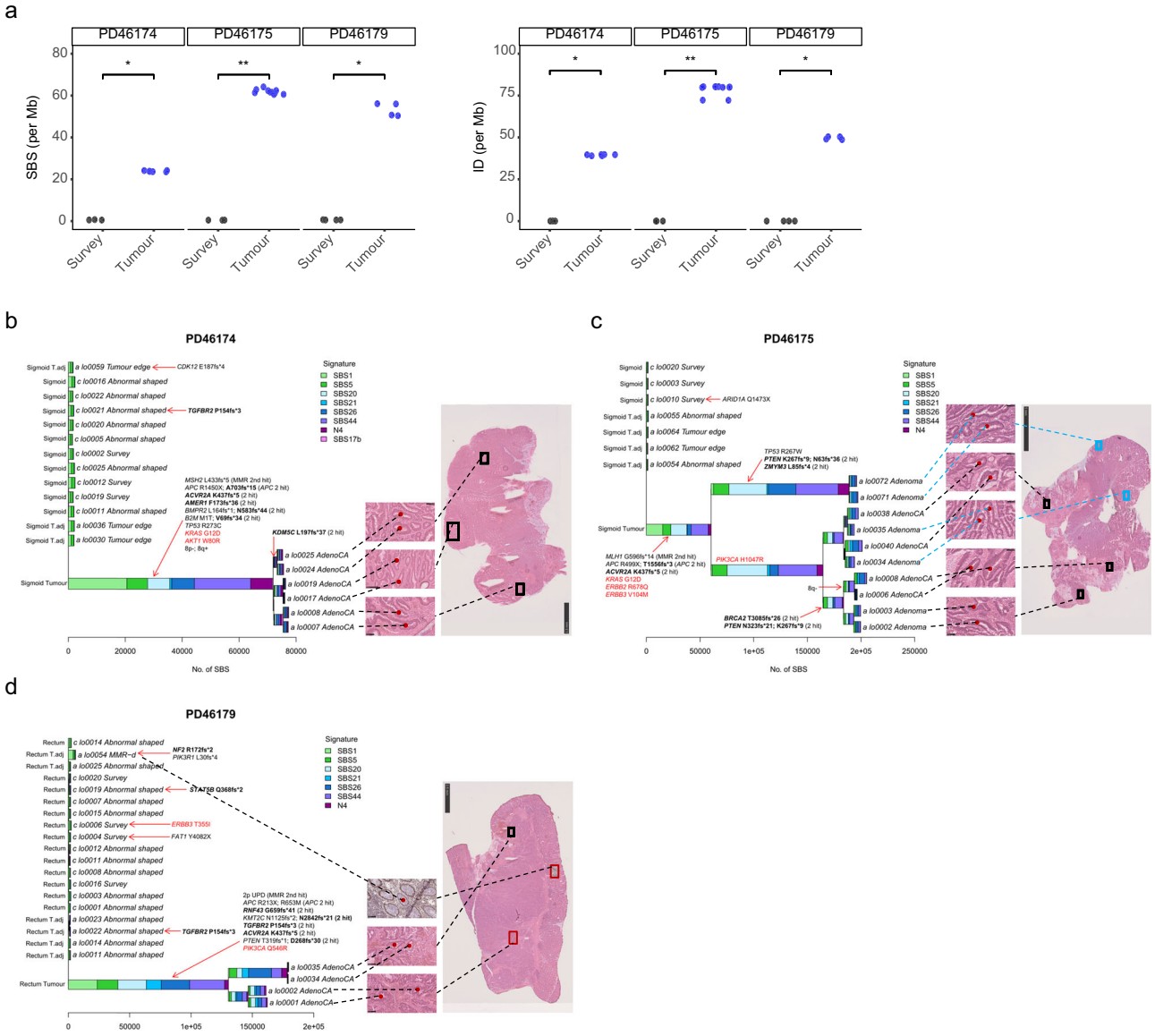

**Fig. 4 SBS mutational signatures in the tumour crypts of LS patients. a** Single base substitutions (SBS) and insertions and deletions (ID) mutation density of tumour crypts (blue) compared to survey crypts (black) in the three patients with a paired colon tumour tissue. Two-sided Wilcoxon test: PD46174 (Survey: n = 3; Tumour: n = 6; p = 0.024), PD46175 (Survey: n = 3; Tumour: n = 10; p = 0.007), PD46179 (Survey: n = 4; Tumour: n = 4; p = 0.029) for both SBS and ID. **b–d** Phylogenetic trees showing the clonal relationship between crypts in the three patients with a paired tumour sample. Branch lengths correspond to the number of SBS mutations. SBS mutational signatures are mapped onto tree branches. Each crypt is annotated with its identifier, crypt type, and tissue origin. Cancer driver mutations and copy number variants are labelled on the corresponding tree branch. Frameshift mutations at microsatellite regions (repeat length ≥ 5) are indicated in bold. Activating mutations of oncogenes are indicated in red. Inactivating mutations of tumour suppressors are indicated in black. For tumour crypts, only the key cancer driver mutations on major tree branches are annotated. Most of the mutations in tumour crypts were frameshift mutations at microsatellite regions, which were consequences of MSI. These mutations were omitted from annotation. The location where each tumour crypt was sampled is shown by the H&E staining of the tumour section to the right of the phylogenetic trees. **c** For PD46175, adenoma crypts were annotated in blue brackets. **d** For PD46179, the inserts of the region within the red brackets show crypts that were sampled from another tissue section. These images of crypts were superimposed onto the tumour section to show its relative position. Scale bar of tumour section: 2.5 mm (PD46174 and PD46179); 5 mm (PD46175).

We compared the mutation burdens in paired normal and tumour ("adenoma" and "adenoCA") crypts from three LS patients in whom these tissues were available. In all three patients, the SBS and ID burdens of tumour crypts were ~100-fold and 2000-fold higher respectively than normal crypts sampled at the same time (Fig. 4a). Mutational signature analysis showed that all tumour crypts displayed SBS1, SBS5, previously described DNA MMR deficiency-associated signatures SBS20, SBS21, SBS26, SBS44, and an unknown signature N4[14,20]. Phylogenetic tree

reconstruction showed that tumour crypts formed a distinct cluster away from normal crypts, and the tumour tree branches arose from a single trunk of at least 50,000 SBSs and multiple driver mutations (Fig. 4b–d). This indicates that many mutations had already accumulated in the most recent common ancestor cell of each tumour after multiple rounds of clonal expansion, with subsequent clonal expansion resulting in the formation of the adenoma or carcinoma. We noticed a greater contribution of SBS1 in tree trunks compared to branches, and MMR

deficiency-associated signatures contributed to most of the mutations in branches. However, the exact mechanism underlying such phenomenon is unknown. In PD46175 an adenocarcinoma evolved from a recent ancestor cell that also evolved to generate an adjacent adenoma (Fig. 4c). In all three tumours, somatic wild type allele inactivation of the relevant MMR gene and APC biallelic mutations were present in all tumour crypts and mapped onto the trunk of the tumour cluster, consistent with existing knowledge of one of the tumourigenesis pathways in LS in which adenoma formation requires both MMR deficiency and APC inactivation[21,22].

As many IDs were generated in the tumour crypts, ID mutational signatures were also extracted in the three patients. All tumour crypts showed mutational signatures ID1, ID2, ID7, and an unknown signature N3 (Supplementary Fig. 4a–c). ID1 and ID2, characterized by single insertions and deletions of thymine at poly-T tracts, respectively, are ubiquitous signatures due to polymerase slippage during DNA replication, with increased exposures observed in samples with MMR deficiency[14,20]. ID7, on the other hand, is characterized by mono- and di-nucleotide deletions at microsatellite regions and is associated with MMR deficiency[14,20]. The unknown signature N3 was characterized by ≥5 bp deletions at sequences with various repeat lengths. Inspection of these deletions revealed that N3 represents polynucleotide deletions at microsatellite regions, presumably due to continual retraction of microsatellites under MMR deficiency (Supplementary Fig. 4e). The morphologically normal, yet MMR deficient crypt (lo0054) of PD46179, showed mutational signatures ID1 and ID2, but not ID7 or N3, suggesting that MMR deficiency was in an early state in which complex IDs had not yet been accumulated in the genome (Supplementary Fig. 4d).

## Discussion

This study presents the genomic landscape of normal epithelial tissues in LS patients. While elevated mutation burdens have been shown in normal tissues from individuals with germline POLE/POLD1 exonuclease domain mutations[6], the same was not observed generally in the epithelium of individuals with germline MMR gene mutations in this study. In fact, the mutation burden of colonic epithelium in LS patients was comparable to that of wild type individuals[10]. Mutation of one allele of the POLE/POLD1 genes appears sufficient to perturb the proofreading function of the DNA polymerase and lead to the accumulation of mutations in the genome, whereas mutation of one allele of an MMR gene in the presence of a wild type allele is insufficient to disrupt the MMR system. The protective function of the wild type MMR allele is further illustrated by a recent study of normal tissues from individuals with constitutional MMR deficiency (CMMRD) due to inherited biallelic mutations in MMR genes, in which the mutation rate of normal intestinal epithelial cells ranged from 500 to 3000 mutations per year, 10 to 60-fold higher than that in LS patients[23].

Despite the substantially greater mutation rate in normal tissues from POLE/POLD1 mutation carriers than from MMR gene mutation carriers, the estimated risk of colorectal cancer by age 70 is only moderately higher in POLE (90%) than in MLH1/MSH2 (41–48%) gene mutation carriers[4,24]. These observations indicate that the elevated mutation rate present in all normal colorectal epithelial cells in POLE/POLD1 mutation carriers leads to a higher rate of development of adenomas compared to wild-type and LS individuals. However, the risk of conversion from adenoma to carcinoma may be greater in MLH1/MSH2 gene mutation carriers, perhaps due to higher ID mutation rates that supervene once the wild-type MMR gene allele is lost in individual cells.

We identified a single morphologically normal but MMR deficient colonic crypt in a young LS patient with an inherited MSH2 mutation. The presence of such crypts in normal colorectum from LS patients has been previously reported. The MMR deficient crypts, as evidenced by loss of MMR protein staining and demonstration of MSI, were found with an estimated frequency of ~1 MMR deficient crypt focus per cm$^2$ of mucosa[25,26]. In this study, we were able to analyse the genome of such a crypt demonstrating the emergence of MMR deficiency-associated mutational signatures and elevated numbers of SBS and ID mutations. This crypt, with its distinctive mutational patterns, may therefore represent an example of a rare, dispersed subpopulation of cells in the intestine of LS patients which are in the earliest stages of progression from normal to cancer cell. If so, given the relatively small number of colorectal neoplasms that develop in LS patients, only a very small minority of such cells with MMR deficiency and elevated mutation rates ever become symptomatic neoplasms. Interestingly, despite thorough interrogation of the whole genome for mutations, CNV or SV, no MSH2 second hit could be identified. This suggests the existence of a state in which mutations accumulate in the crypt and MSI starts to emerge before the MMR gene second hit takes place. Indeed, this state may contribute to the emergence of fully defective MMR function by contributing the second hit. The biological mechanism underlying this state is unclear though. Nevertheless, we do not exclude the possibility that the absence of MMR gene second hit was merely due to undetected mutations/inactivating mechanisms given the analysis pipeline and technology platform used in this study.

Based on the phylogenetic tree reconstruction on tumour crypts, we showed that MMR deficiency and APC inactivation are the earliest events in LS tumour progression, with hotspot mutations in other cancer genes such as those in KRAS and TP53 serving as late cancer drivers. While we could not delineate the sequence of mutations within a trunk of mutations, 2 of the 3 tumours had APC frameshift mutations at microsatellite regions, possibly hinting that APC inactivation came after MMR deficiency in the 2 tumours. Overall, our findings are consistent with the pre-existing tumourigenesis pathways described in LS[21,22].

In conclusion, this study demonstrates that the overwhelming majority of normal epithelial crypts with germline heterozygous MMR gene mutations are genomically stable. In future work, larger scale systematic screening for and genetic profiling of the small subpopulation of MMR deficient crypts in the normal epithelium of LS patients will clarify the earliest molecular events a normal colorectal stem cell requires in order to acquire MMR deficiency and subsequently progress to neoplastic transformation.

## Methods

**Human tissue samples**. A total of 10 LS patients were included in this study. Normal epithelial tissues from these patients were obtained at the time of tumour resection surgery or from a rectal biopsy. The epithelial tissues were of various origins, including gastric (PD45540), endometrial (PD45541), and intestinal (PD43853, PD45539, PD46174, PD46175, PD46176, PD46177, PD46178 and PD46179) origin. A paired colorectal tumour specimen was available in 3 patients (PD46174, PD46175, PD46179). Except for one patient (PD43853), peripheral blood samples were obtained. All samples were collected with informed consent from the patients. The study was approved by the IRB of the University of Hong Kong/ Hospital Authority Hong Kong West Cluster (UW14-257) and the UK National Health Service (NHS) Research Ethics Committee (REC 17/WM/0295 and 15/WA/0131).

**Sample preparation**. Frozen tissues were embedded in optimal cutting temperature (OCT) compound and cryo-sectioned at 25 μm on PEN-membrane slides and fixed in 70% ethanol for 5 min. Fixed tissue sections were rinsed twice with 1X PBS, stained with haematoxylin and eosin (H&E), and photo-recorded using a Nanozoomer S60 Slide scanner (Hamamatsu). Crypts of interest were marked using the

NDP.View2 software (Hamamatsu) for subsequent laser-capture microdissection (LCM).

**Immunohistochemistry (IHC).** Frozen tissues were cryo-sectioned at 6 μm and immediately fixed in absolute ethanol for 10 min. The sections were rinsed in 1X TBS for 5 min. IHC staining was performed following the protocol from the ImmPRESS™ Excel HRP Staining Kit, Anti-Mouse IgG, Peroxidase (Vector Laboratories #MP-7602-15). Primary antibody incubation was performed at 4 °C overnight with the following dilutions: 1:50 MLH1 (G168-15) (BD Pharmingen™ #551092); 1:100 MSH2 (Ab2) (Millipore #NA27); 1:100 PMS2 (BD Pharmingen™ #556415). MMR protein antibodies were used according to the germline MMR gene mutation in respective patients. The sections were counter-stained with haematoxylin. In the case where the MMR deficient crypt was found, the immediate consecutive 25 μm section was chosen for subsequent LCM.

**Crypt type classification.** Crypts were classified into 7 types according to their morphologies and MMR protein staining patterns. 1. Survey: canonical crypts with normal crypt architecture lined by orderly differentiated cells; 2. Specific morphologies: crypts with budding, branching, or unusual shapes; 3. MMR-deficient (MMR-d): morphologically normal crypts with MMR protein loss; 4. Intestinal metaplasia (IM): gastric crypts with proliferative crypt base and goblet cells; 5. Tumour edge: phenotypically normal crypts immediately adjacent to the tumour; 6. Adenoma: tumour crypts from the recognizable remnant adenoma area with dysplastic cells lining the adenomatous glands; 7. Adenocarcinoma (Ade-noCA): invasive tumour glands lying in the desmoplastic stroma.

**Laser capture microdissection.** Crypts of interest were microdissected using an LMD7000 compound laser microdissection microscope (Leica) and collected in skirted 96 well plates. Crypts were lysed in Proteinase K/Reconstitution buffer using PicoPure® DNA Extraction Kit (Arcturus®). DNA extraction was done at 65 °C for 3 h, followed by 75 °C for 30 min, and kept at −20 °C before library preparation.

**Library preparation and whole genome sequencing.** Library preparation adapted for low DNA input (100–1000 cells) was performed as described previously[10,12,27]. Briefly, DNA was enzymatically fragmented and A-tailed using NEBNext® Ultra™ II FS DNA Library Prep Kit (NEB). After adaptor ligation, DNA was amplified for 12 PCR cycles using KAPA HiFi HotStart ReadyMix (KAPA Biosystems). DNA libraries with concentration > 13 ng/μL were deemed to be successful and subjected to Pair-End 151 whole genome sequencing at 30X using NovaSeq 6000 (Illumina).

**Single base substitutions calling and variant filtering.** Mutation calling and strategy for variant filtering were performed as described previously[10,12]. Briefly, sequencing reads were aligned to the human reference genome version GRCh37 (hg19) using Burrow-Wheeler Aligner (BWA) and duplicate reads were removed[28]. Single base substitutions (SBS) were called using the Cancer Variants through Expectation Maximization (CaVEMan) algorithm, with major and minor copy number specified at 5 and 2, respectively[29]. Variants were called against an unmatched normal to retain early embryonic mutations, which was crucial for later phylogenetic tree reconstruction. A set of inhouse post-processing filters[27] were applied to remove (1) common single nucleotide polymorphisms (SNPs) using a panel of 75 unmatched normal samples; (2) mapping artefacts, in which only variants with supporting reads with a median alignment score ASMD ≥ 140 and fewer than half of the reads were clipped (CLPM = 0) were retained; (3) over-lapping reads, as a result of small insert size; (4) LCM-specific artefacts arising from cruciform DNA formation during the library preparation.

Variants were then genotyped in all the samples from each patient using an inhouse algorithm (cgpVAF). The number of mutant and wild-type reads in every sample over every site in each patient were concatenated. Only reads with mapping quality ≥ 30 and base quality ≥ 25 were counted. Using samples without copy number variants, a one-sided exact binomial test was applied in each patient to remove germline variants. Variants were further fitted in a beta-binomial distribution and the overdispersion factor (rho) for each variant was estimated using maximum likelihood estimation. Rho values < 0.1 were filtered out. This removed artefactual variants that were distributed across samples with low depths, but wrongly captured in the previous exact binomial test. Detailed code for the removal of germline variants can be found at GitHub (https://github.com/TimCoorens/Unmatched_NormSeq).

**Phylogenetic tree reconstruction.** For each sample, variants with VAF > 0.3 and <0.1 were marked as present ("1") and absent ("0"), respectively. Variants with VAF between 0.1 and 0.3 were marked as ambiguous ("?"). Based on this assig-nation, a phylogenetic tree was built in each patient using the maximum parsimony method[30], with 1000 bootstrapping. Variants were then assigned to tree branches using a maximum likelihood approach. Owing to the small number of insertions and deletions (INDELs) in normal samples, trees built from SBSs were used to assign INDELs to branches for higher accuracy.

**INDEL calling.** INDELs were called using the Pindel algorithm[31,32]. Similar to SBS calling, an unmatched normal was used. Variants were filtered for having sum of mapping quality of supporting reads Qual ≥ 300 and total read depth ≥ 15. Genotyping and germline variant removal were performed as described above, except a threshold of rho < 0.2 was used in the beta-binomial filter.

**Mutational signature extraction and assignment.** SBS variants were assigned to 96 categories based on their trinucleotide contexts, as described in the "Mutational Signature" working group of the Pan-Cancer Analysis of Whole Genomes (PCAWG) study[20]. Mutational signature extraction was performed using the Hierarchical Dirichlet Process (HDP) method (https://github.com/nicolaroberts/hdp)[33]. HDP uses a nonparametric Bayesian approach for mutational signature analysis, with the benefits of simultaneously matching to known signatures (priors) and extracting de novo signatures, as well as automatically learning the number of signatures needed to explain the data. To correctly infer mutagenic processes operating at different times during crypt development, each branch of the phylo-genetic tree was treated as a sample. To balance extraction accuracy and efficiency, only branches with ≥100 mutations were included, and branch length was restricted to 2500 mutations. A hierarchy with patients as parent nodes and tree branches as child notes was constructed. No reference signature was used as priors. Gibbs sampling of posterior distribution was run with 80,000 burn-in iterations, sampling 2500 iterations with 250 spacing and 3 concentration parameters. Extraction was restricted, in which components with cosine similarity ≥0.9 were merged (cos.merge = 0.9) and significant exposure was present in at least 2 sam-ples (min.sample = 2). The initial extraction yielded 10 components (Supple-mentary Fig. 5a). Using an expectation-maximization algorithm, extracted components were deconvoluted into suspected reference signatures based on cosine similarity prediction and biological relevance (Supplementary Fig. 5b). Composite signatures with a contribution >0.15 were reconstituted and tested for cosine similarity with the original component (Supplementary Fig. 5c). A cosine similarity >0.9 was considered as a good reconstitution. As the decomposed signatures failed to reconstitute and represent component N4 (cosine similarity = 0.85), potential composite signatures SBS15, SBS20, SBS26, and SBS44 were subtracted from component N4, and the residue was treated as an unknown N4 signature (Sup-plementary Fig. 5a). Following deconvolution, mutational signatures, from which the component exposure >0.1, were reassigned and fitted to tree branches using sigfit (https://github.com/kgori/sigfit)[34].

INDEL (ID) mutational signature extraction was performed as described above. 5 components were extracted (Supplementary Fig. 6a). After deconvolution, composite signatures with a contribution >0.2 were reconstituted (Supplementary Fig. 6b, c). As component N3 was poorly reconstituted (cosine similarity = 0.83), potential composite signatures ID1 and ID2 were subtracted from component N3 and the residue was treated as an unknown N3 signature (Supplementary Fig. 6a). To reduce contamination of ID1 and ID2 in component N4, which was virtually signature ID18, ID1 and ID2 were subtracted from it before signature reassignment and fitting to tree branches.

**Mutational signature validation.** Mutational signature was also extracted using SigProfiler, a computational tool which uses a non-random matrix factorization (NMF)-based method for signature extraction[20]. Extraction was done using 15 end processes, 50 total iterations and genome build GRCh37. De novo signatures extracted with the most stable solution (6 SBS signatures and 4 ID signatures) were compared to components extracted by HDP (Supplementary Fig. 7a, b). Signatures extracted by SigProfiler were less resolved but were all accounted for by compo-nents of HDP (Supplementary Fig. 7c, d).

**Cancer driver discovery.** A list of 369 driver genes that has been implicated in human cancers was used to filter the variants called by CaVEMan and Pindel[35]. Variants were filtered for coding, non-synonymous and protein-truncating muta-tions, followed by driver classification using Cancer Genome Interpreter (https://www.cancergenomeinterpreter.org/home). Variants classified as "known", and truncating variants in tumour suppressors classified as "predicted", were con-sidered as driver mutations. Missense variants classified as "predicted" were filtered for cancer hotspots using MutationMapper[36,37] (https://www.cbioportal.org/mutation_mapper). Variants reported as "hotspots" or "predicted oncogenic" were considered as driver mutations. Cancer driver mutations were reviewed and con-firmed in JBrowse[38].

**Copy number variants calling.** Copy number variants were called using the Allele-Specific Copy number Analysis of Tumours (ASCAT) algorithm[39,40]. Matched blood samples were used as controls. For patient PD43853 for which no blood sample was available, a matched colonic crypt was used as a control for the duo-denum, ileum, and appendix samples, and a matched ileum crypt was used as a control for the colon samples. To reduce false positive calls and noise, a segmen-tation penalty of 150 was applied in all samples. Variants ≤2 Mb in length, present in the sex chromosomes or at centromeric regions were removed. Positive calls were manually reviewed and confirmed using JBrowse.

**Structural variants calling**. Structural variants were called using the Breakpoints via Assembly (BRASS) algorithm (https://github.com/cancerit/BRASS). Same as calling copy number variants, matched blood samples or a distant clone from the same individual were used as controls. Variants were filtered using an inhouse algorithm, AnnotateBRASS (https://github.com/MathijsSanders/AnnotateBRASS), which utilizes a list of statistics and a panel of normal samples for the removal of false positive calls[12]. Calls absent in the corresponding control and supported by sufficient read pairs (brass score > 10) were considered as true variants. Positive calls were reviewed and confirmed using JBrowse.

**Clonal organoid culture**. Normal ileum, caecum, and descending colon epithelial tissues from PD45539, as well as normal gastric body and antrum tissues from PD45540 were used to establish organoid cultures, as described previously[41]. As a control, organoid was also developed using the normal sigmoid colon tissue from a non-Lynch 40-year-old female patient. Organoids were sorted into single cells, and individual ASCs were grown and hand-picked for clonal organoid expansion (each clonal organoid was designated as a parent clone). For each tissue origin, 2–4 parent clonal organoids were expanded for 6 weeks or less (to minimize somatic mutation gain during the post-sort culturing period) and DNA for whole genome sequencing were sampled. To assess the impact of in vitro culture on LS organoids, these parent clonal organoids were further cultured for 4–6 months. During this period, each ASC acquired new mutations independently, making the clonal organoid heterogeneous over time. A second round of clone picking was performed after the long-term culture, where 2 daughter clonal organoids were hand-picked after cell sorting from each parent clone, grown short term for expansion, and whole genome sequenced. While the parent clones harboured somatic mutations accumulated in vivo prior to long term culture, the daughter clones contained both somatic mutations accumulated in vivo prior to long-term culture and in vitro-induced mutations acquired during the long-term culturing period (Supplementary Fig. 3a).

**Whole genome sequencing of organoids**. Genomic DNA extraction was done according to the instructions of the AllPrep DNA/RNA/miRNA Universal Kit manual (QIAGEN). DNA libraries were prepared based on the manufacturer's protocol of the KAPA Hyper Prep Kit (Roche). Briefly, 500 ng of genomic DNA was mechanically fragmented using the Covaris S2 ultrasonicator. Following end-repair and A-tailing of the DNA, IDT xGen® Dual Indexed UMI adaptors were ligated to the DNA. Adaptor-ligated libraries were size-selected in the range of 300–750 bp using the dual-SPRI size selection method and were amplified with 5 PCR cycles. The libraries were validated with the Agilent Bioanalyzer, quantified using Qubit and qPCR assays before loading to the NovaSeq 6000 (Illumina) for Pair-End 151 bp sequencing at 30X.

**Variant filtering and signature extraction for organoids**. Sequencing reads were mapped against the reference genome GRCh37 (hg19) using Burrow-Wheeler Aligner (BWA) and SAMtools[28,42]. Variants were called using HaplotypeCaller with genotyping mode "DISCOVERY" and stand_call_conf of 50. SNVs with Total Depth (DP) < 8 and Fisher Strand Score (FS) > 60 were filtered out. INDELs with DP < 8 and FS > 200 were filtered out. Variants kept after filtering were annotated using ANNOVAR[43]. The somatic variants in the parental clonal organoids and in vitro-induced mutations in the corresponding daughter clonal organoids were obtained through a variant filtering strategy similar to that previously described by Blokzijl et al.[44]. Briefly, common SNPs with population frequency ≥ 0.01 in "1000g2015aug_all" or "1000g2015aug_eas" databases were removed. For somatic variant detection in the parent clones, germline variants were removed based on their presence in the corresponding blood samples. Those somatic variants with insufficient coverage (DP ≤ 20) in either the parent clonal organoid or blood sample were also removed. Finally, somatic variants with low allele fraction (VAF ≤ 0.3) were removed. This removed any variants after the first division of the sorted ASCs. Variant filtering for daughter clones was similar except that the corresponding parent clones, instead of blood samples, were used as references. This removed the in vivo somatic mutations that were inherited from the corresponding parent clone, leaving only the mutations acquired in vitro during the long-term culturing period. Mutational signature extraction and assignment for organoids was done as described above, except that reference signatures, namely SBS1, SBS2, SBS3, SBS5, SBS13, SBS16, SBS17a, SBS17b, SBS18, SBS25 SBS28, SBS30, SBS37, SBS40, SBS41, SBS43, SBS45, and SBS49 were used as priors.

**Reporting summary**. Further information on research design is available in the Nature Research Reporting Summary linked to this article.

## Data availability
Raw whole genome sequencing data are deposited and available in the European Genome Phenome Archive (EGA) with accession number EGAD00001008092. The guidelines for patient consent prevent the derived data files from being dispersed by open access. To ensure the data is used for academic and research purposes, controlled access of the data will be available indefinitely upon request made to the WTSI CGP Data access committee. The remaining data are available within the Supplementary Data in this paper.

## Code availability
Code for mutation calling is available through GitHub (https://github.com/cancerit). Code for variant filtering and tree building is available through GitHub (https://github.com/TimCoorens/Unmatched_NormSeq). Code for mutational signature extraction is available upon request.

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

## Acknowledgements

We thank the staff of the Wellcome Sanger Institute Sample Logistics, Sequencing, and Informatics facilities for their contribution to this study. We thank Dorothy H.T. Cheng, Laura O'Neill, Yvette Hooks, Stephen Gamble, Calli Latimer, and Kirsty Roberts for their support with patient coordination, sample management and laboratory work, and clinicians in Hong Kong Hospital Authority for clinical care. This work was supported by a Cancer Research UK Grand Challenge Award [C98/A24032], the Wellcome Trust, the Kadoorie Charitable Foundation, the Hong Kong Cancer Fund, a Health and Medical Research Fund from the Food and Health Bureau, The Government of the Hong Kong Special Administrative Region (Project No. 04151366) and a theme-based research grant from the Research Grants Council of the Hong Kong Special Administrative Region, China (Project No. T12-710/16R). We thank the Genomics Core and the Imaging and Flow cytometry Core of the Centre for PanorOmic Sciences (CPOS), The University of Hong Kong for providing organoid whole genome sequencing and flow cytometry services.

## Author contributions

B.C.H.L., P.S.R., S.Y.L., and M.R.S. designed the study. B.C.H.L. and J.H. performed the crypt sequencing laboratory work. B.C.H.L., S.S.K.Y., W.Y.T., and H.H.N.Y. performed organoid work. B.C.H.L., P.S.R., T.H.H.C., H.L.-S., and M.A.S. contributed to statistical analyses and variant filtering. B.C.H.L. and H.C.S. performed bioinformatics analysis of organoid culture. H.L.-S. and S.O. provided data from control cohort and assisted model testing. S.J.A.B, A.S.Y.C., A.K.W.C., S.L.H., S.T.Y., and S.Y.L. coordinated patient data and samples. B.C.H.L. and P.S.R. performed data analysis. B.C.H.L., S.Y.L., and M.R.S. wrote the manuscript with input and assistance from all authors. S.Y.L. and M.R.S. supervised the study.

## Competing interests

All authors declare no competing interests.
