## [Peer review file · Nature Communications]

REVIEWER COMMENTS

Reviewer #1 (Remarks to the Author): Expert in Lynch Syndrome and colorectal cancer genetics and genomics

Lee et al reports the genomic features, measured by WGS, of 132 non-neoplastic crypts from 10 patients with Lynch syndrome. The crypts were classified into 7 subtypes based on morphology, MMR protein expression and relation to adenocarcinoma. The number of single base substitutions and indel somatic mutations were measured in each crypt and compared with previously measured crypts from non-Lynch syndrome. Mutational signatures were determined to ascertain evidence of defect MMR within each crypt.

Morphologically normal crypts from LS patients were not significantly different from non-LS crypts. An exception to this was a single crypt that showed loss of MSH2 expression and genomic evidence of MMR-deficiency similar to crypts within adenocarcinoma. This is an important study and helps to understand the early steps in tumorigenesis in Lynch syndrome. The study would have benefited from the analysis of additional MMR-deficient crypts to provide a more comprehensive view of this important precursor in the tumorigenesis process.

comments:

1. can the authors explain why some of the crypts show SBS18, a signature associated with defective base excision repair caused by germline biallelic MUTYH pathogenic variants?

2. the study only evaluated a single MMR-deficient crypt although the text in the discussion hypothesizes a new state in the tumorigenesis process..."the existence of a state in which mutations accumulate in the crypt and MSI starts to emerge before the MMR gene second hit takes place." This statement seems quite a stretch based on the absence of identifying a second hit in the single crypt examined. The failure to identify a second hit within a Lynch related CRC is not uncommon and yet biallelic inactivation of the MMR gene is the established mechanism. suggest the authors reword this statement to reflect the limitation/s of their analysis.

3. In recent years multiple pathways of tumorigenesis have been postulated in Lynch syndrome CRC (see PMID: 29424427). given the potential importance of the development of interval CRCs from MMR-deficient crypt pathway, the authors should comment on how their results might fit within this proposed model (or not) and with APC, CTNNB1, KRAS and TP53 mutations.

4. there is a predominance of crypts analysed that are from the distal colon and rectum yet Lynch-related CRCs are more likely to occur in the proximal colon. did the authors see any difference in SBS

and ID mutations by anatomical site in the colon and rectum? this may need to be discussed as a potential limitation?

minor comment

1. page 7, line 307...."into below" needs editing

congratulations on your study.

Reviewer #2 (Remarks to the Author): Expert in colorectal cancer organoids and functional genomics

The manuscript by Lee et al. identifies the mutational landscape in normal cells of Lynch Syndrome patients. This disease is the most common cause of hereditary colorectal cancer and affects a significant percentage of patients. The authors use whole genome sequencing to demonstrate that the mutational rate of these cells do not change significantly and that are genomically stable. They report that only one single crypt shows signs of early MMR deficiency.

The work is original as the mutational landscape by NGS has not been previously reported in normal cells of Lynch syndrome patients. The findings may not have a huge impact to the field, however, it is an interesting study that was needed to be done and important to be reported to the field. The group has an excellent background in NGS and identifying mutational landscapes of different cell types (Moore et al Nature, Lee-Six et al., Cell and many other manuscripts).

The results are well interpreted, and the authors discuss well the findings. The methodology is based on the innovative technology NGS (the group is a referent in the field). There are enough details provided in the methods section. It is obvious that the team has a broad experience on this technology.

Major points:

My main concern is the low number of samples analyzed. Previous publications as for example Lee-Six et al., identifying normal colorectal epithelial cells used 42 individuals while here are only 10 patients samples.

Minor points:

1. Graphs in general are too small. It would be better to make them bigger (and increase font size). I would suggest to create figure 4 instead of leaving 3 figures.
2. I understand that the authors use in their experiments these names (PD45539...) but is difficult to follow. It should be changed to simple names PD1, PD2, PD3..
3. References from Halazonetis group are missing. This group has published interesting results on normal and colorectal cancer cells.
4. Last three sentences of the abstract are not informative. Should be changed to explain better the findings
"
5. It should be written in the abstract how many individuals were used to do this study.

Reviewer #3 (Remarks to the Author): Expert in cancer predisposition, Lynch Syndrome genetics and genomics

Lee and coworkers present a study in which they present the somatic mutational processes in crypts of normal epithelial tissues from Lynch syndrome patients in comparison to healthy controls and MMR-deficient tumors. The study reveals that the mutational burden and processes in normal crypts is not different between LS and non-LS patients, suggesting that (as could have expected) complete MMR deficiency is required to accelerate pathogenicity. The use of crypts to enable clonal assessment of normal cells is elegant and effective and the results are interesting and relevant. I have some comments I would like to ask the authors to respond to:

1. A single crypt is described that shows loss of MSH2 protein expression in IHC, suggesting a second hit, but based on a thorough search the conclusion was drawn that a second hit somatic mutation in the wild type MSH2 allele is absent. This has led to a rather challenging hypothesis that there could be a degree of MMR deficiency, leading to increased mutation rates, even before the second hit mutation occurs. It may indeed be that such an intermediate state exists, but with an observation in one crypt, this hypothesis appears rather speculative. What makes the authors so certain that there is no mutation in the wild type allele that was missed with this strategy, like e.g. methylation or a hidden genomic aberration? Couldn't there be an alternative scenario in which no additional hit is required for pathogenesis? Can the authors provide more evidence that other MMR deficient lesions in normal crypts following the same route?

2. It is remarkable to see that this MMR-d lo0054 lesion carries a relatively low contribution of MMR-associated mutations compared to the tumors, and that most of the mutational load is caused by the typical clock-like mutational processes. In neoplastic lesions the contribution of clock-like signatures is higher in the trunk than in the branches, suggesting that these typical MMR-associated signatures arise as the dominant mutational processes only later during tumorigenesis. Could the authors elaborate on this in their manuscript? Is the initial rise in mutations only due to increased proliferation, and why is this pattern changing in time?

3. SBS88, associated with colibactin exposure, was found to be present in every crypt from two siblings. This is interesting as it may contribute to our knowledge on the contribution of such genotoxins to the risk of cancer. Are these individuals at increased risk of developing malignancies? Can the authors say something about the relative contribution of this signature to what is observed in tumours. It appears lower, but a proper analysis might be needed here.

4. A subset of the normal crypts in LS individuals also contained driver mutations. Please indicate whether the number and type of driver mutations was different between normal cells in LS patients and non-LS patients.

5. Are microsatellites instable in the normal and tumor crypts or are they clonal?

We thank the reviewers for their constructive comments.

Below please find our point-to-point response to the Reviewers' comment and amendment to the manuscript. Apart from the amendment related to addressing the comments of the reviewers, we have also made minor editing in various parts of the manuscript, including addition of a paragraph summarizing the key findings of the study at the end of the introduction (line 55), to conform with the format of *Nature Communications*.

Reviewer #1 (Remarks to the Author): Expert in Lynch Syndrome and colorectal cancer genetics and genomics

Lee et al reports the genomic features, measured by WGS, of 132 non-neoplastic crypts from 10 patients with Lynch syndrome. The crypts were classified into 7 subtypes based on morphology, MMR protein expression and relation to adenocarcinoma. The number of single base substitutions and indel somatic mutations were measured in each crypt and compared with previously measured crypts from non-Lynch syndrome. Mutational signatures were determined to ascertain evidence of defect MMR within each crypt.

Morphologically normal crypts from LS patients were not significantly different from non-LS crypts. An exception to this was a single crypt that showed loss of MSH2 expression and genomic evidence of MMR-deficiency similar to crypts within adenocarcinoma. This is an important study and helps to understand the early steps in tumorigenesis in Lynch syndrome. The study would have benefited from the analysis of additional MMR-deficient crypts to provide a more comprehensive view of this important precursor in the tumorigenesis process.

comments:

1. can the authors explain why some of the crypts show SBS18, a signature associated with defective base excision repair caused by germline biallelic *MUTYH* pathogenic variants?

SBS18 is a mutational signature associated with DNA damage caused by reactive oxygen species. This signature is usually seen in some (but not all) gastrointestinal crypts, reflecting the oxidative environment (i.e. gastrointestinal tract) which the crypts inhabit. SBS18 is characterized with distinctive C>A changes. However, it should not be confused with SBS36, a signature associated with defective base excision repair and has its own distinctive C>A patterns in the 96 mutation type categories. Moreover, based on our WGS data, none of the crypts examined contain pathogenic *MUTYH* mutations which would suggest the presence of defective base excision repair.

2. the study only evaluated a single MMR-deficient crypt although the text in the discussion hypothesizes a new state in the tumorigenesis process..."the existence of a state in which mutations accumulate in the crypt and MSI starts to emerge before the MMR gene second hit takes place." This statement seems quite a stretch based on the absence of identifying a second hit in the single crypt examined. The failure to identify a second hit within a Lynch related CRC is not uncommon and yet biallelic inactivation of the MMR gene is the established mechanism. suggest the authors reword this statement to reflect the limitation/s of their analysis.

We thank the reviewer for this point. The failure to identify a second hit in LS-associated CRC in conventional sequencing can sometimes be attributed to insufficient sequencing depth in a bulk sample with low tumour purity. Here, given the fact that WGS of the MMR deficient crypt (a clonal unit) achieved a sequencing depth of 35X, we are reasonably confident that there is no conventional second hit at the genomic level. However, we certainly cannot exclude the possibility of epigenetic inactivation of the second *MSH2* allele (e.g. by *MSH2* promoter methylation) as a second hit.

Based on our experience, an LS cancer patient inherited with a germline genetic *MSH2* mutation is always accompanied with a somatic genetic second hit or loss of heterozygosity (LOH) in the tumour. Indeed, we speculate that epigenetic changes (e.g. aberrant methylation) on the wild type *MSH2* allele might be the reason for the early, elevated mutations and MSI in the MMR deficient crypt. These epigenetic changes, however, only act on the *MSH2* wild type allele transiently. Hence, we proposed the presence of a stage in crypts where mutations accumulate and MSI emerges before a fixed mutation at the genetic level constitutes a stable second hit and fully inactivates the MMR system.

We do agree with the Reviewer, however, that such a hypothesis is based on a few assumptions and based on limited data. To avoid overstating, we have added a sentence in line 257, describing the possibility that we might have just missed an undiscovered second hit given the current analysis pipeline and technology platform.

3. In recent years multiple pathways of tumorigenesis have been postulated in Lynch syndrome CRC (see PMID: 29424427). given the potential importance of the development of interval CRCs from MMR-deficient crypt pathway, the authors should comment on how their results might fit within this proposed model (or not) and with APC, CTNNB1, KRAS and TP53 mutations.

We thank the reviewer for the suggestion. We have added a paragraph in the Discussion (line 260) discussing how our findings relate to the current proposed tumorigenesis pathways in LS.

We did not discuss about LS tumours in great details as we wanted to focus our discussion on the stable mutation rate of normal cells in LS, which is the core thesis of

this manuscript. However, we would like to respond to the reviewer/anyone who might be interested in the LS tumours as below:

Based on the phylogenetic tree reconstruction, MMR second hit mutations and *APC* 2 hit mutations are mapped to the long trunk of mutations in all the 3 tumours studied. This means MMR deficiency and *APC* inactivation are the earliest genetic events in driving progression, in particular clonal expansion of the tumour. This is consistent with the proposed pathways of LS tumourigenesis where normal mucosa develops into adenoma which then progresses to carcinoma. However, our analysis could not further delineate the sequence of mutations within a trunk of mutations, hence we were not able to determine which of the two processes (i.e. MMR deficiency and *APC* inactivation) happened first before the other in the 3 tumours studied.

According to our clinical data, all 3 adenocarcinomas were not interval cancers, and they do not contain *CTNNB1* activating mutations. This is consistent with the reported mutual exclusivity of *APC* and *CTNNB1* mutations in WNT signaling activation in CRC.

From the phylogenetic trees, *KRAS* and *TP53* mutations were late cancer drivers. This is evidenced by the presence of *KRAS* activating mutations in 2 (PD46174 and PD46175) out of 3 tumour trunks, and the presence of a *TP53* inactivating mutation in the tumour sub-branch in PD46175. These late cancer drivers contribute to the progression and development of tumours.

All in all, the findings in this study agree with the pre-existing tumourigenesis pathways described in LS.

4. there is a predominance of crypts analysed that are from the distal colon and rectum yet Lynch-related CRCs are more likely to occur in the proximal colon. did the authors see any difference in SBS and ID mutations by anatomical site in the colon and rectum? This may need to be discussed as a potential limitation?

Sporadic CRCs with MSI occur predominantly in the proximal colon. However, in LS, the frequencies of CRC in the proximal and distal colon are similar. We did not see any difference in SBS or ID mutation load in the colon and rectum by anatomical site or by proximity, hence we pooled all the colonic crypts as a single group for simplicity and clarity.

minor comment

1. page 7, line 307...."into below" needs editing

Thank you very much for spotting the typographical error. We corrected it accordingly.

congratulations on your study.

Reviewer #2 (Remarks to the Author): Expert in colorectal cancer organoids and functional genomics

The manuscript by Lee et al. identifies the mutational landscape in normal cells of Lynch Syndrome patients. This disease is the most common cause of hereditary colorectal cancer and affects a significant percentage of patients. The authors use whole genome sequencing to demonstrate that the mutational rate of these cells do not change significantly and that are genomically stable. They report that only one single crypt shows signs of early MMR deficiency.

The work is original as the mutational landscape by NGS has not been previously reported in normal cells of Lynch syndrome patients. The findings may not have a huge impact to the field, however, it is an interesting study that was needed to be done and important to be reported to the field. The group has an excellent background in NGS and identifying mutational landscapes of different cell types (Moore et al Nature, Lee-Six et al., Cell and many other manuscripts).

The results are well interpreted, and the authors discuss well the findings. The methodology is based on the innovative technology NGS (the group is a referent in the field). There are enough details provided in the methods section. It is obvious that the team has a broad experience on this technology.

Major points:

My main concern is the low number of samples analyzed. Previous publications as for example Lee-Six et al., identifying normal colorectal epithelial cells used 42 individuals while here are only 10 patients samples.

We take the point and are aware of the relatively small sample size in this compared to the Lee-Six study of normal colon in normal individuals and individuals with sporadic colorectal cancer, and we agree that increasing both the patient and crypt number would have provided additional confidence in drawing conclusions. However, although LS is the most common hereditary CRC type, it accounts for only 3% of all CRCs and thus availability of the types of samples necessary for this type of work is limited. Despite this, we managed to obtain epithelial tissues from 10 LS patients, with mutations in the different LS predisposing genes and from the various organs (i.e. the stomach, the colon and the endometrium) that are commonly affected in LS. Moreover, during microdissection of normal crypts, we minimized sampling bias by selecting crypts that are histologically distant from each other. On balance, we believe that we have made a reasonable effort in the crypt surveying process, and that our data represent the genomic landscape of epithelial tissues in LS objectively.

Minor points:

1. Graphs in general are too small. It would be better to make them bigger (and increase font size). I would suggest to create figure 4 instead of leaving 3 figures.

Thank you very much for the comment. We apologize for the small fonts and figures. We arranged the figures in a way (putting figures and legends on the same page) that complies with the formatting guidelines of Nature Communications.

Taking the reviewer's advice, we split Figure 1 into 2 separate figures, and enlarged some panels in Figure 2 (became Figure 3 after the edit). We hoped the amendments give better presentation.

2. I understand that the authors use in their experiments these names (PD45539...) but is difficult to follow. It should be changed to simple names PD1, PD2, PD3..

We understand it might be difficult for readers to follow long patient codes while reading the manuscript. However, changing the patient code would make it extremely hard for us to track the data. To avoid mixing up data and for easy future referencing, we would like to stick to the original labelling.

3. References from Halazonetis group are missing. This group has published interesting results on normal and colorectal cancer cells.

We did not cite some of the papers published by the Halazonetis group because most of their work has involved mouse models with various genetically modified backgrounds. While their results are doubtless interesting and illuminating, we find it less appropriate to make a direct comparison with our study which focuses on human tissues with a LS background. For example, a paper published by Lugli *et al.* in Cell Reports in 2017 involved the use of $Apc^{min/+}$ mice for mutation rate analysis. Such findings would be highly relevant to groups studying mutational landscape of tissues in Familial Adenomatous Polyposis (FAP) individuals.

4. Last three sentences of the abstract are not informative. Should be changed to explain better the findings

We tried hard to comply with the abstract word limit while conveying the findings of our study. We apologize for the compact wordings. We rephrased the sentences and hoped that the abstract is more comprehensible to readers.

5. It should be written in the abstract how many individuals were used to do this study.

Thank you for the suggestion. We added the number of individuals examined in the abstract accordingly.

Reviewer #3 (Remarks to the Author): Expert in cancer predisposition, Lynch Syndrome genetics and genomics

Lee and coworkers present a study in which they present the somatic mutational processes in crypts of normal epithelial tissues from Lynch syndrome patients in comparison to healthy controls and MMR-deficient tumors. The study reveals that the mutational burden and processes in normal crypts is not different between LS and non-LS patients, suggesting that (as could have expected) complete MMR deficiency is required to accelerate pathogenicity. The use of crypts to enable clonal assessment of normal cells is elegant and effective and the results are interesting and relevant. I have some comments I would like to ask the authors to respond to:

1. A single crypt is described that shows loss of MSH2 protein expression in IHC, suggesting a second hit, but based on a thorough search the conclusion was drawn that a second hit somatic mutation in the wild type MSH2 allele is absent. This has led to a rather challenging hypothesis that there could be a degree of MMR deficiency, leading to increased mutation rates, even before the second hit mutation occurs. It may indeed be that such an intermediate state exists, but with an observation in one crypt, this hypothesis appears rather speculative. What makes the authors so certain that there is no mutation in the wild type allele that was missed with this strategy, like e.g. methylation or a hidden genomic aberration? Couldn't there be an alternative scenario in which no additional hit is required for pathogenesis? Can the authors provide more evidence that other MMR deficient lesions in normal crypts following the same route?

We performed WGS on the MMR deficient crypt with a sequencing depth of 35X. We did not find any mutations on the *MSH2* wild type allele that would suggest a pathogenic variant. The WGS approach cannot exclude the possibility of having aberrant methylation or other epigenetic changes on the *MSH2* wild type allele being a second hit. However, based on our experience, an LS cancer patient with a germline *MSH2* mutation is always accompanied with a somatic genetic second hit or loss of heterozygosity (LOH) in the tumour. Indeed, we speculate that epigenetic changes (e.g. aberrant methylation) on the wild type *MSH2* allele might be the reason for the early, elevated mutations and MSI in the MMR deficient crypt. These epigenetic changes, however, only act on the *MSH2* wild type allele transiently, and they serve as initiating events for the mutator phenotype and contribute to the later second hit in the crypt. We believe that only after a second hit is installed genetically in a crypt can it become a full-fledged MMR deficient crypt and spark off pathogenesis.

Unfortunately, in all of the studies reporting the observation of MMR-deficient crypt foci, none of them were able to perform comprehensive survey to document if a second hit was found in those crypt foci. Given the difficulty in finding such crypts and concurrently being able to microdissect them for WGS, our study is the first in the field to be able to study one of them. We think a follow up mechanistic investigation of

MMR deficiency crypt foci in the LS tumorigenesis process would be of great value to the field.

Similar to the response to Reviewer 1, to avoid overstating, we added a sentence describing the possibility that we might have just missed an undiscovered second hit given the current analysis pipeline and technology platform (line 257).

2. It is remarkable to see that this MMR-d lo0054 lesion carries a relatively low contribution of MMR-associated mutations compared to the tumors, and that most of the mutational load is caused by the typical clock-like mutational processes. In neoplastic lesions the contribution of clock-like signatures is higher in the trunk than in the branches, suggesting that these typical MMR-associated signatures arise as the dominant mutational processes only later during tumorigenesis. Could the authors elaborate on this in their manuscript? Is the initial rise in mutations only due to increased proliferation, and why is this pattern changing in time?

Yes, the reviewer is right in spotting the clock-like signature SBS1 being a dominant mutational signature during the early development of tumour crypts, as evidenced by the high SBS1 contribution in the MMR deficient crypt (PD46179_lo0054) and greater SBS1 exposure in the tumour trunk compared to branches.

We do not know the exact mechanism leading to the shift in the contribution of SBS1 and MMR-associated signatures from the trunk to branches. However, it is conceivable that the abrogation of G:T mismatch surveillance is the most imminent consequence of MMR dysfunction. This is because an MMR-deficient cell has just transited from a normal state where ageing (through deamination of 5-methylcytosine) is the main mutagenic process. An increased mitotic rate of MMR-deficient cells during early tumorigenesis might be another reason. Nevertheless, it appears that additional events are required for other MMR-associated signatures (i.e. SBS20, SBS21, SBS26 and SBS44) to happen in a cell. The nature of these additional events is currently unknown though.

We have added a sentence in line 193 describing this phenomenon.

3. SBS88, associated with colibactin exposure, was found to be present in every crypt from two siblings. This is interesting as it may contribute to our knowledge on the contribution of such genotoxins to the risk of cancer. Are these individuals at increased risk of developing malignancies? Can the authors say something about the relative contribution of this signature to what is observed in tumours. It appears lower, but a proper analysis might be needed here.

These two siblings were healthy LS gene carriers at relatively young ages, thus it is not possible to document an increased cancer risk. Based on our data, SBS88 can exist in the normal colon of both LS and non-LS individuals. This mutational signature contributes to

extra mutagenesis in the colon, but its contribution to CRCs is low (enriched in 2-5% of CRCs according to Pleguezuelos-Manzano *et al.* (Nature, 2020)). Besides, Dziubańska-Kusibab *et al.* (Nature Medicine, 2020) studied the patterns of mutagenesis generated by colibactin specifically in CRCs. Using WGS data from 200 CRCs, they estimated that mutations in the colibactin damage motif (CDM) constitutes on average 0.5% of total mutations in CRCs.

Although we did not have enough samples to investigate the contribution of SBS88 specifically in LS tumours, we think that the results would be similar to sporadic CRCs in general.

4. A subset of the normal crypts in LS individuals also contained driver mutations. Please indicate whether the number and type of driver mutations was different between normal cells in LS patients and non-LS patients.

Since the number of drivers observed were small, it would be difficult to perform statistical analysis on the relative numbers and types between LS and normal individuals as statistical fluctuation may begin to play a role. However, based on the result we have generated, there does not appear to be a substantial difference in driver compositions between LS patients and normal individuals overall.

5. Are microsatellites instable in the normal and tumor crypts or are they clonal?

All INDELS in both normal and tumour crypts were clonal, as demonstrated by the median variant allele frequency (VAF) > 0.3. Restricting INDELS at microsatellite regions gave the same results. Given the fact that INDEL mutation rate and ID mutation spectrum of normal crypts between LS and non-LS individuals are highly similar, we confirm that the microsatellite regions are stable in the normal crypts of LS individuals. On the other hand, the ID mutation spectrum of tumour crypts showed mutational signatures ID7 and N3, together with a higher contribution of ID2 compared to ID1, suggesting the microsatellite regions are unstable in the tumour crypts.

REVIEWERS' COMMENTS

Reviewer #1 (Remarks to the Author):

Thank you to the authors for addressing my comments and revising the manuscript accordingly.

Reviewer #2 (Remarks to the Author):

Thanks for your comments.

The authors believe that the number of samples analyzed are enough. I don't really agree but as it is now written in the abstract, the reader will know this limitation.

I still think that these long codes to name the patients are not helping to follow the results.

The manuscript has been improved and in my opinion can be accepted for publication in Nature communications.

Reviewer #3 (Remarks to the Author):

I thank the authors for their clear rebuttal, I have no further comments

Reviewer #1 (Remarks to the Author):

Thank you to the authors for addressing my comments and revising the manuscript accordingly.

We thank the reviewer for the advice along the reviewing process.

Reviewer #2 (Remarks to the Author):

Thanks for your comments.

The authors believe that the number of samples analyzed are enough. I don't really agree but as it is now written in the abstract, the reader will know this limitation.

I still think that these long codes to name the patients are not helping to follow the results.

The manuscript has been improved and in my opinion can be accepted for publication in Nature communications.

We believe the coherent results from both the crypt data and organoid data will ease the concerns over limited sample size. We thank the reviewer for the understanding on the use of original patient codes. We are grateful for the reviewer's comments.

Reviewer #3 (Remarks to the Author):

I thank the authors for their clear rebuttal, I have no further comments

We thank the reviewer for the positive feedback and suggestions in the reviewing process.